# MEMORY AUGMENTED CONTROL NETWORKS

**Arbaaz Khan, Clark Zhang, Nikolay Atanasov, Konstantinos Karydis,
Vijay Kumar, Daniel D. Lee**

**GRASP Laboratory, University of Pennsylvania**

## ABSTRACT

Planning problems in partially observable environments cannot be solved directly with convolutional networks and require some form of memory. But, even memory networks with sophisticated addressing schemes are unable to learn intelligent reasoning satisfactorily due to the complexity of simultaneously learning to access memory and plan. To mitigate these challenges we propose the Memory Augmented Control Network (MACN). The network splits planning into a hierarchical process. At a lower level, it learns to plan in a locally observed space. At a higher level, it uses a collection of policies computed on locally observed spaces to learn an optimal plan in the global environment it is operating in. The performance of the network is evaluated on path planning tasks in environments in the presence of simple and complex obstacles and in addition, is tested for its ability to generalize to new environments not seen in the training set.

## 1 INTRODUCTION

A planning task in a partially observable environment involves two steps: inferring the environment structure from local observation and acting based on the current environment estimate. In the past, such *perception-action* loops have been learned using supervised learning with deep networks as well as deep reinforcement learning (Daftry et al., 2016), (Chebotar et al., 2016), (Lee et al., 2017). Popular approaches in this spirit are often end-to-end (i.e. mapping sensor readings directly to motion commands) and manage to solve problems in which the underlying dynamics of the environment or the agent are too complex to model. Approaches to learn end-to-end perception-action loops have been extended to complex reinforcement learning tasks such as learning how to play Atari games (Mnih et al., 2013a), as well as to imitation learning tasks like controlling a robot arm (Levine et al., 2015).

Purely convolutional architectures (CNNs) perform poorly when applied to planning problems due to the reactive nature of the policies learned by them (Zhang et al., 2016b), (Giusti et al., 2016). The complexity of this problem is compounded when the environment is only partially observable as is the case with most real world tasks. In planning problems, when using a function approximator such as a convolutional neural network, the optimal actions are dependent on an internal state. If one wishes to use a state-less network (such as a CNN) to obtain the optimal action, the input for the network should be the whole history of observations and actions. Since this does not scale well, we need a network that has an internal state such as a recurrent neural network or a memory network. (Zhang et al., 2016a) showed that when learning how to plan in partially observable environments, it becomes necessary to use memory to retain information about states visited in the past. Using recurrent networks to store past information and learn optimal control has been explored before in (Levine, 2013). While (Siegelmann & Sontag, 1995) have shown that recurrent networks are Turing complete and are hence capable of generating any arbitrary sequence in theory, this does not always translate into practice. Recent advances in memory augmented networks have shown that it is beneficial to use external memory with read and write operators that can be learned by a neural network over recurrent neural networks (Graves et al., 2014), (Graves et al., 2016). Specifically, we are interested in the Differentiable Neural Computer (DNC) (Graves et al., 2016) which uses an external memory and a network controller to learn how to read, write and access locations in the external memory. The DNC is structured such that computation and memory operations are separated from each other. Such a memory network can in principle be plugged into the convolutional

architectures described above, and be trained end to end since the read and write operations are differentiable. However, as we show in our work, directly using such a memory scheme with CNNs performs poorly for partially observable planning problems and also does not generalize well to new environments.

To address the aforementioned challenges we propose the *Memory Augmented Control Network (MACN)*, a novel architecture specifically designed to learn how to plan in partially observable environments under sparse rewards.[1] Environments with sparse rewards are harder to navigate since there is no immediate feedback. The intuition behind this architecture is that planning problem can be split into two levels of hierarchy. At a lower level, a planning module computes optimal policies using a feature rich representation of the locally observed environment. This local policy along with a sparse feature representation of the partially observed environment is part of the optimal solution in the global environment. Thus, the key to our approach is using a planning module to output a local policy which is used to augment the neural memory to produce an optimal policy for the global environment. Our work builds on the idea of introducing options for planning and knowledge representation while learning control policies in MDPs (Sutton et al., 1999). The ability of the proposed model is evaluated by its ability to learn policies (continuous and discrete) when trained in environments with the presence of simple and complex obstacles. Further, the model is evaluated on its ability to generalize to environments and situations not seen in the training set.

The key contributions of this paper are:

1. A new network architecture that uses a differentiable memory scheme to maintain an estimate of the environment geometry and a hierarchical planning scheme to learn how to plan paths to the goal.

2. Experimentation to analyze the ability of the architecture to learn how to plan and generalize in environments with high dimensional state and action spaces.

## 2 METHODOLOGY

Section 2.1 outlines notation and formally states the problem considered in this paper. Section 2.2 and 2.3 briefly cover the theory behind value iteration networks and memory augmented networks. Finally, in section 2.4 the intuition and the computation graph is explained for the practical implementation of the model.

### 2.1 PRELIMINARIES

Consider an agent with state $s_t \in \mathcal{S}$ at discrete time $t$. Let the states $\mathcal{S}$ be a discrete set $[s_1, s_2, \ldots, s_n]$. For a given action $a_t \in \mathcal{A}$, the agent evolves according to known deterministic dynamics: $s_{t+1} = f(s_t, a_t)$. The agent operates in an unknown environment and must remain safe by avoiding collisions. Let $m \in \{-1, 0\}^n$ be a *hidden* labeling of the states into free $(0)$ and occupied $(-1)$. The agent has access to a sensor that reveals the labeling of nearby states through an observations $z_t = H(s_t)m \in \{-1, 0\}^n$, where $H(s) \in \mathbb{R}^{n \times n}$ captures the local field of view of the agent at state $s$. The local observation consists of ones for observable states and zeros for unobservable states. The observation $z_t$ contains zeros for unobservable states. Note that $m$ and $z_t$ are $n \times 1$ vectors and can be indexed by the state $s_t$. The agent's task is to reach a goal region $\mathcal{S}^{\text{goal}} \subset \mathcal{S}$, which is assumed obstacle-free, i.e., $m[s] = 0$ for all $s \in \mathcal{S}^{\text{goal}}$. The information available to the agent at time $t$ to compute its action $a_t$ is $h_t := (s_{0:t}, z_{0:t}, a_{0:t-1}, \mathcal{S}^{\text{goal}}) \in \mathcal{H}$, where $\mathcal{H}$ is the set of possible sequences of observations, states, and actions. Our problem can then be stated as follows :

**Problem 1.** *Given an initial state $s_0 \in \mathcal{S}$ with $m[s_0] = 0$ (obstacle-free) and a goal region $\mathcal{S}^{goal}$, find a function $\mu : \mathcal{S} \to \mathcal{A}$ such that applying the actions $a_t := \mu(s_t)$ results in a sequence of states $s_0, s_1, \ldots, s_T$ satisfying $s_T \in \mathcal{S}^{goal}$ and $m[s_t] = 0$ for all $t = 0, \ldots, T$.*

Instead of trying to estimate the hidden labeling $m$ using a mapping approach, our goal is to learn a policy $\mu$ that maps the sequence of sensor observations $z_0, z_1, \ldots z_T$ directly to actions for the agent. The partial observability requires an explicit consideration of *memory* in order to learn $\mu$ successfully. A partially observable problem can be represented via a Markov Decision Process (MDP) over

---

[1] In this work an agent receives a reward only when it has reached the goal prescribed by the planning task.

the history space $\mathcal{H}$. More precisely, we consider a finite-horizon discounted MDP defined by $\mathcal{M}(\mathcal{H}, \mathcal{A}, \mathcal{T}, r, \gamma)$, where $\gamma \in (0, 1]$ is a discount factor, $\mathcal{T} : \mathcal{H} \times \mathcal{A} \to \mathcal{H}$ is a deterministic transition function, and $r : \mathcal{H} \to \mathbb{R}$ is the reward function, defined as follows:

$$\mathcal{T}(h_t, a_t) = (h_t, s_{t+1} = f(s_t, a_t), z_{t+1} = H(s_{t+1})m, a_t)$$
$$r(h_t, a_t) = z_t[s_t]$$

The reward function definition stipulates that the reward of a state $s$ can be measured only after its occupancy state has been observed.

Given observations $z_{0:t}$, we can obtain an estimate $\hat{m} = \max\{\sum_\tau z_\tau, -1\}$ of the map of the environment and use it to formulate a locally valid, fully-observable problem as the MDP $\mathcal{M}_t(\mathcal{S}, \mathcal{A}, f, r, \gamma)$ with transition function given by the agent dynamics $f$ and reward $r(s_t) := \hat{m}[s_t]$ given by the map estimate $\hat{m}$.

## 2.2 VALUE ITERATION NETWORKS

The typical algorithm to solve an MDP is Value Iteration (VI) (Sutton & Barto, 1998). The value of a state (i.e. the expected reward over the time horizon if an optimal policy is followed) is computed iteratively by calculating an action value function $Q(s, a)$ for each state. The value for state $s$ can then be calculated by $V(s) := \max_a Q(s, a)$. By iterating multiple times over all states and all actions possible in each state, we can get a policy $\pi = \arg\max_a Q(s, a)$. Given a transition function $\mathcal{T}_r(s'|s, a)$, the update rule for value iteration is given by (1)

$$V_{k+1}(s) = \max_a [r(s, a) + \gamma \sum_{s'} \mathcal{T}_r(s'|s, a)V_k(s)] \ . \tag{1}$$

A key aspect of our network is the inclusion of this network component that can approximate this Value Iteration algorithm. To this end we use the VI module in Value Iteration Networks (VIN) (Tamar et al., 2016). Their insight is that value iteration can be approximated by a convolutional network with max pooling. The standard form for windowed convolution is

$$V(x) = \sum_{k=x-w}^{x+w} V(k)u(k) \ . \tag{2}$$

(Tamar et al., 2016) show that the summation in (2) is analogous to $\sum_{s'} \mathcal{T}(s'|s, a)V_k(s)$ in (1). When (2) is stacked with reward, max pooled and repeated K times, the convolutional architecture can be used to represent an approximation of the value iteration algorithm over K iterations.

## 2.3 EXTERNAL MEMORY NETWORKS

Recent works on deep learning employ neural networks with external memory (Graves et al., 2014), (Graves et al., 2016), (Kurach et al., 2015), (Parisotto & Salakhutdinov, 2017). Contrary to earlier works that explored the idea of the network learning how to read and access externally fixed memories, these recent works focus on learning to read and write to external memories, and thus forgo the task of designing what to store in the external memory. We are specifically interested in the DNC (Graves et al., 2016) architecture. This is similar to the work introduced by (Oh et al., 2016) and (Chen et al., 2017). The external memory uses differentiable attention mechanisms to determine the degree to which each location in the external memory $M$ is involved in a read or write operation. The DNC makes use of a controller (a recurrent neural network such as LSTM) to learn to read and write to the memory matrix. A brief overview of the read and write operations follows.[2]

### 2.3.1 READ AND WRITE OPERATION

The read operation is defined as a weighted average over the contents of the locations in the memory. This produces a set of vectors defined as the read vectors. $R$ read weightings $\{w_t^{read,1}, \ldots, w_t^{read,R}\}$ are used to compute weighted averages of the contents of the locations in the memory. At time $t$ the read vectors $\{re_t^1, \ldots, re_t^R\}$ are defined as :

$$re_t^i = M_t^\top w_t^{read,i} \tag{3}$$

---

[2]We refer the interested reader to the original paper (Graves et al., 2016) for a complete description.

where $w_t^{read,i}$ are the read weightings, $re_t$ is the read vector, and $M_t$ is the state of the memory at time $t$. These read vectors are appended to the controller input at the next time step which provides it access to the memory. The write operation consists of a write weight $w_t^W$, an erase vector $e_t$ and a write vector $v_t$. The write vector and the erase vector are emitted by the controller. These three components modify the memory at time $t$ as :

$$M_t = M_{t-1}(1 - w_t^W e_t^\top) + w_t^W v_t^\top \ . \tag{4}$$

Memory addressing is defined separately for writing and reading. A combination of content-based addressing and dynamic memory allocation determines memory write locations, while a combination of content-based addressing and temporal memory linkage is used to determine read locations.

## 2.4 Memory Augmented Control Model

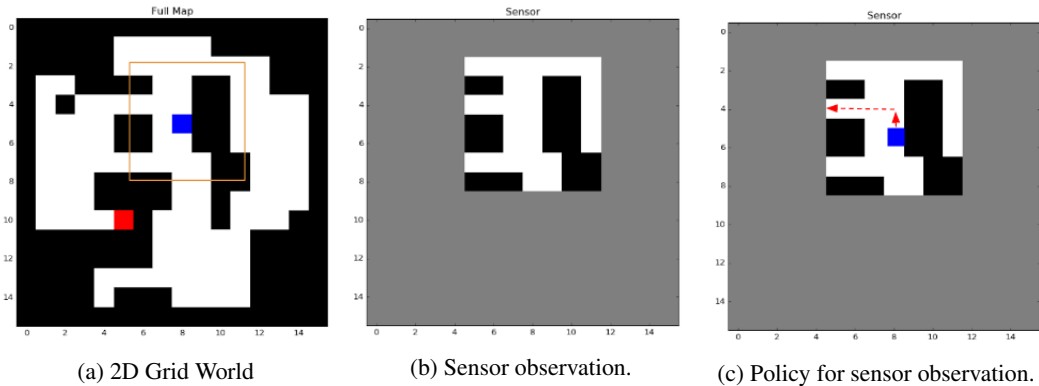

(a) 2D Grid World     (b) Sensor observation.     (c) Policy for sensor observation.

Figure 1: **2D Environment a)** Let the agent (blue square) operate in a 2D environment. The goal region is represented by the red square and the orange square represents the agents observation **b)** Agents observation. The gray area is not observable. **c)** It is possible to plan on this locally observed space since it is a MDP.

Consider the 2D grid world in Fig 1a. The agent is spawned randomly in this world and is represented by the blue square. The goal of the agent is to learn how to navigate to the red goal region. Let this environment in Fig 1a be represented by a MDP $\mathcal{M}$. The key intuition behind designing this architecture is that planning in $\mathcal{M}$ can be decomposed into two levels. At a lower level, planning is done in a local space within the boundaries of our locally observed environment space. Let this locally observed space be $z'$. Fig 1b represents this locally observed space. As stated before in Section 2.1, this observation can be formulated as a fully observable problem $\mathcal{M}_t(\mathcal{S}, \mathcal{A}, f, r, \gamma)$. It is possible to plan in $\mathcal{M}_t$ and calculate the optimal policy for this local space, $\pi_l^*$ independent of previous observations (Fig 1c). It is then possible to use any planning algorithm to calculate the optimal value function $V_l^*$ from the optimal policy $\pi_l^*$ in $z'$. Let $\Pi = [\pi_l^1, \pi_l^2, \pi_l^3, \pi_l^4, \ldots, \pi_l^n]$ be the list of optimal policies calculated from such consecutive observation spaces $[z_0, z_1, \ldots z_T]$. Given these two lists, it is possible to train a convolutional neural network with supervised learning. The network could then be used to compute a policy $\pi_l^{new}$ when a new observation $z^{new}$ is recorded.

This policy learned by the convolutional network is purely reactive as it is computed for the $z^{new}$ observation independent of the previous observations. Such an approach fails when there are local minima in the environment. In a 2D/3D world, these local minima could be long narrow tunnels culminating in dead ends (see Fig 2). In the scenario where the environment is populated with tunnels, (Fig 2) the environment is only partially observable and the agent has no prior knowledge about the structure of this tunnel forcing it to explore the tunnel all the way to the end. Further, when entering and exiting such a structure, the agent's observations are the same, i.e $z_1 = z_2$, but the optimal actions under the policies $\pi_l^1$ and $\pi_l^2$ (computed by the convolutional network) at these time steps are not the same, i.e $a_{\pi^1} \neq a_{\pi^2}$. To backtrack successfully from these tunnels/nodes, information about previously visited states is required, necessitating memory.

To solve this problem, we propose using a differentiable memory to estimate the map of the environment $\hat{m}$. The controller in the memory network learns to selectively read and write information

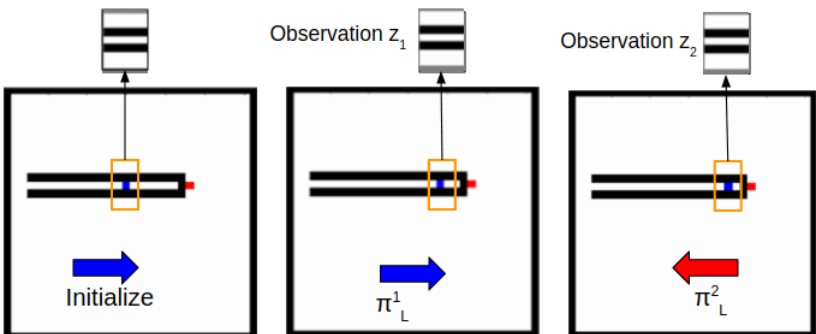

Figure 2: **Environment with local minima.** The agents observation when entering the tunnel to explore it and when backtracking after seeing the dead end are the same. Using a reactive policy for such environments leads to the agent getting stuck near the dead end .

to the memory bank. When such a differentiable memory scheme is trained it is seen that it keeps track of important events/landmarks (in the case of tunnel, this is the observation that the dead end has been reached) in its memory state and discards redundant information. In theory one can use a CNN to extract features from the observation $z'$ and pass these features to the differentiable memory. Instead, we propose the use of a VI module (Tamar et al., 2016) that approximates the value iteration algorithm within the framework of a neural network to learn value maps from the local information. We hypothesize that using these value maps in the differential memory scheme provides us with better planning as compared to when only using features extracted from a CNN. This architecture is shown in Figure 3.

The VI module is setup to learn how to plan on the local observations $z$. The local value maps (which can be used to calculate local policies) are concatenated with a low level feature representation of the environment and sent to a controller network. The controller network interfaces with the memory through an access module (another network layer) and emits read heads, write heads and access heads. In addition, the controller network also performs its own computation for planning. The output from the controller network and the access module are concatenated and sent through a linear layer to produce an action. This entire architecture is then trained end to end. Thus, to summarize, the planning problem is solved by decomposing it into a two level problem. At a lower level a feature rich representation of the environment (obtained from the current observation) is used to generate local policies. At the next level, a representation of the histories that is learned and stored in the memory, and a sparse feature representation of the currently observed environment is used to generate a policy optimal in the global environment.

**Computation Graph**: To explain the computation graph, consider the case of a 2D grid world with randomly placed obstacles, a start region and a goal region as shown in Fig 1a. The actions for this grid world are considered to be discrete. The 2D grid world is presented in the form of an image $I$ of size $m \times n$ to the network. Let the goal region be $[m_{goal}, n_{goal}]$ and the start position be $[m_{start}, n_{start}]$. At any given instant, only a small part of $I$ is observed by the network and the rest of the image $I$ is blacked out. This corresponds to the agent only observing what is visible within the range of its sensor. In addition to this the image is stacked with a reward map $R_m$ as explained in (Tamar et al., 2016). The reward map consists of an array of size $m \times n$ where all elements of the array except the one corresponding to index $[m_{goal}, n_{goal}]$ are zero. Array element corresponding to $[m_{goal}, n_{goal}]$ is set to a high value(in our experiments it is set to 1) denoting reward. The input image of dimension $[m \times n \times 2]$ is first convolved with a kernel of size $(3 \times 3)$, 150 channels and stride of 1 everywhere. This is then convolved again with a kernel of size $(1,1)$, 4 channels and stride of 1. Let this be the reward layer $R$. $R$ is convolved with another filter of size $(3,3)$ with 4 channels. This is the initial estimate of the action value function or $Q(s, a)$. The initial value of the state $V(s)$ is also calculated by taking max over $Q(s, a)$. The operations up to this point are summarized by the "Conv" block in Figure 3. Once these initial values have been computed, the model executes a for loop k times (the value of k ranges based on the task). Inside the for loop at every iteration, the R and V are first concatenated. This is then convolved with another filter of size (3,3) and 4 channels to get the updated action value of the state, $Q(s, a)$. We find the value of the state V(s) by taking the max

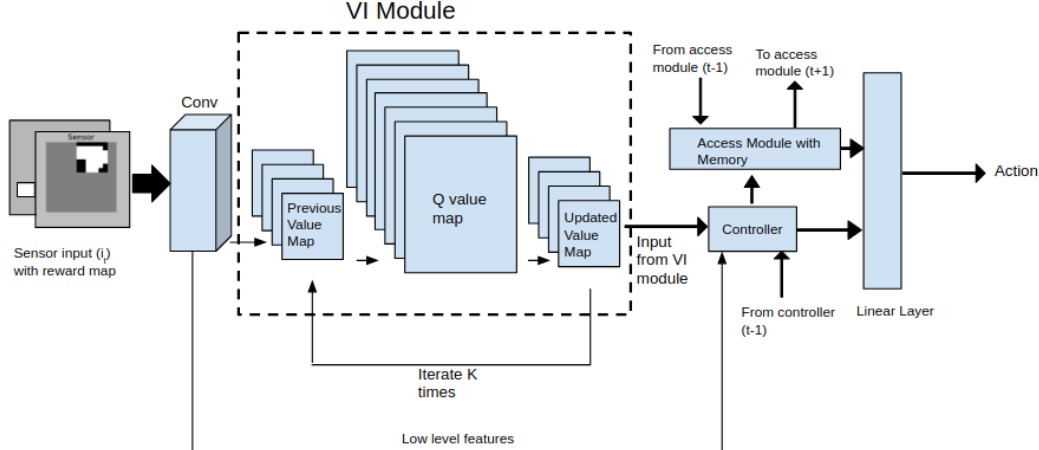

Figure 3: **MACN Architecture.** The architecture proposed uses convolutional layers to extract features from the environment. The value maps are generated with these features. The controller network uses the value maps and low level features to emit read and write heads in addition to doing its own planning computation.

of the action value function. The values of the kernel sizes are constant across all three experiments. The updated value maps are then fed into a DNC controller. The DNC controller is a LSTM (hidden units vary according to task) that has access to an external memory. The external memory has 32 slots with word size 8 and we use 4 read heads and 1 write head. This varies from task to task since some of the more complex environments need more memory. The output from the DNC controller and the memory is concatenated through a linear layer to get prediction for the action that the agent should execute. The optimizer used is the RMSProp and we use a learning rate of 0.0001 for our experiments.

This formulation is easy enough to be extended to environments where the state space is larger than two dimensions and the action space is larger. We demonstrate this in our experiments.

## 3 EXPERIMENTS

To investigate the performance of MACN, we design our experiments to answer three key questions:

- Can it learn how to plan in partially observable environments with sparse rewards?
- How well does it generalize to new unknown environments?
- Can it be extended to other domains?

We first demonstrate that MACN can learn how to plan in a 2D grid world environment. Without loss of generality, we set the probability of all actions equal. The action space is discrete, $\mathcal{A} :=\{down, right, up, left\}$. This can be easily extended to continuous domains since our networks output is a probability over actions. We show this in experiment 3.4. We then demonstrate that our network can learn how to plan even when the states are not constrained to a two dimensional space and the action space is larger than four actions.

### 3.1 NAVIGATION IN PRESENCE OF SIMPLE OBSTACLES

We first evaluate the ability of our network to successfully navigate a 2D grid world populated with obstacles at random positions. We make the task harder by having random start and goal positions. The full map shown in Fig. 4 is the top down view of the entire environment. The input to the network is the sensor map, where the area that lies outside the agents sensing abilities is grayed out as explained before. **VIN**: With just the VI module and no memory in place, we test the performance of the value iteration network on this 2D partially observable environment. **CNN + Memory**: We setup

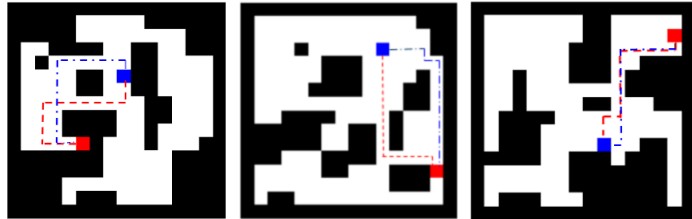

Figure 4: **Performance on grid world environment** . In the map the blue square represents the start position while the red square represents the goal. The red dotted line is the ground truth and the blue dash line represents the agents path. The maps shown here are from the test set and have not been seen before by the agent during training.

| Model | Performance | $16 \times 16$ | $32 \times 32$ | $64 \times 64$ |
|---|---|---|---|---|
| VIN | *Success(%)* | 0 | 0 | 0 |
| | *Test Error* | 0.63 | 0.78 | 0.81 |
| CNN + Memory | *Success(%)* | 0.12 | 0 | 0 |
| | *Test Error* | 0.43 | 0.618 | 0.73 |
| MACN (LSTM) | *Success (%)* | 88.12 | 73.4 | 64 |
| | *Test Error* | 0.077 | 0.12 | 0.21 |
| MACN | *Success(%)* | **96.3** | **85.91** | **78.44** |
| | *Test Error* | **0.02** | **0.08** | **0.13** |

Table 1: **Performance on 2D grid world with simple obstacles**: All models are tested on maps generated via the same random process, and were not present in the training set. Episodes over $40$ (for a $16 \times 16$ wide map), 60 (for $32 \times 32$) and 80 (for $64 \times 64$) time steps were terminated and counted as a failure. Episodes where the agent collided with an obstacle were also counted as failures.

a CNN architecture where the sensor image with the reward map is forward propagated through four convolutional layers to extract features. We test if these features alone are enough for the memory to navigate the 2D grid world. A natural question to ask at this point is can we achieve planning in partially observable environments with just a planning module and a simple recurrent neural network such as a LSTM. To answer this we also test **MACN with a LSTM** in place of the memory scheme. We present our results in Table 1. These results are obtained from testing on a held out test-set consisting of maps with random start, goal and obstacle positions.

Our results show that MACN can learn how to navigate partially observable 2D unknown environments. Note that the VIN does not work by itself since it has no memory to help it remember past actions. We would also like to point out that while the CNN + Memory architecture is similar to (Oh et al., 2016), its performance in our experiments is very poor due to the sparse rewards structure. MACN significantly outperforms all other architectures. Furthermore, MACN's drop in testing accuracy as the grid world scales up is not as large compared to the other architectures. While these results seem promising, in the next section we extend the experiment to determine whether MACN actually learns how to plan or it is overfitting.

## 3.2 NAVIGATION IN PRESENCE OF LOCAL MINIMA

The previous experiment shows that MACN can learn to plan in 2D partially observable environments. While the claim that the network can plan on environments it has not seen before stands, this is weak evidence in support of the generalizability of the network. In our previous experiment the test environments have the same dimensions as in the training set, the number of combinations of random obstacles especially in the smaller environments is not very high and during testing some of the wrong actions can still carry the agent to the goal. Thus, our network could be overfitting and may not generalize to new environments. In the following experiment we test our proposed network's capacity to generalize.

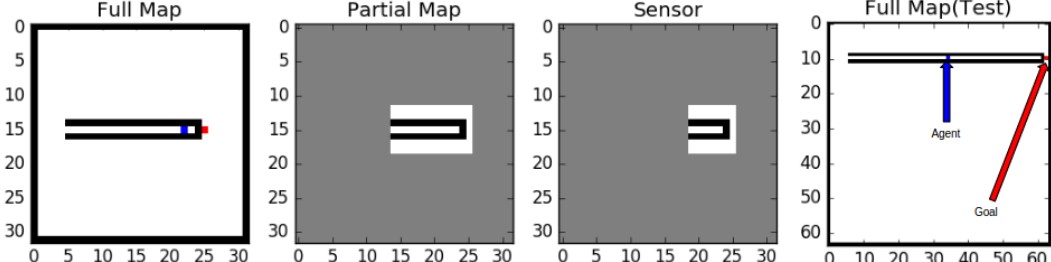

Figure 5: **Grid world environment with local minima**. **Left:** In the full map the blue square represents the current position while the red square represents the goal. **Center-left:** The partial map represents a stitched together version of all states explored by the agent. Since the agent does not know if the tunnel culminates in a dead end, it must explore it all the way to the end. **Center-right:** The sensor input is the information available to the agent. **Right:** The full map that we test our agent on after being trained on smaller maps. The dimensions of the map as well as the tunnel are larger.

The environment is setup with tunnels. The agent starts off at random positions inside the tunnel. While the orientation of the tunnel is fixed, its position is not. To comment on the the ability of our network to generalize to new environments with the same task, we look to answer the following question: *When trained to reach the goal on tunnels of a fixed length, can the network generalize to longer tunnels in bigger maps not seen in the training set?*

The network is set up the same way as before. The task here highlights the significance of using memory in a planning network. The agent's observations when exploring the tunnel and exiting the tunnel are the same but the actions mapped to these observations are different. The memory in our network remembers past information and previously executed policies in those states, to output the right action. We report our results in Table 2. To show that traditional deep reinforcement learning performs poorly on this task, we implement the DQN architecture as introduced in (Mnih et al., 2013b). We observe that even after one million iterations, the DQN does not converge to the optimal policy on the training set. This can be attributed to the sparse reward structure of the environment. We report similar findings when tested with A3C as introduced in (Mnih et al., 2016). We also observe that the CNN + memory scheme learns to turn around at a fixed length and does not explore the longer tunnels in the test set all the way to the end.

| Model | Success (%) | Maximum generalization length |
|---|---|---|
| DQN | 0 | 0 |
| A3C | 0 | 0 |
| CNN + Memory | 12 | 20 |
| VIN | 0 | 0 |
| **MACN** | **100** | **330** |

Table 2: **Performance on grid world with local minima**: All models are trained on tunnels of length 20 units. The success percentages represent the number of times the robot reaches the goal position in the test set after exploring the tunnel all the way to the end. Maximum generalization length is the length of the longest tunnel that the robot is able to successfully navigate after being trained on tunnels of length 20 units.

These results offer insight into the ability of MACN to generalize to new environments. Our network is found capable of planning in environments it has not seen in the training set at all. On visualizing the memory (see supplemental material), we observe that there is a big shift in the memory states only when the agent sees the end of the wall and when the agent exits the tunnel. A t-sne (Maaten & Hinton, 2008) visualization over the action spaces (see Fig. 6) clearly shows that the output of our network is separable. We can conclude from this that the network has learned the spatial structure of the tunnel, and it is now able to generalize to tunnels of longer length in larger maps.

Thus, we can claim that our proposed model is generalizable to new environments that are structurally similar to the environments seen in the training set but have not been trained on. In addition to this in all our experiments are state and action spaces have been constrained to a small number of dimensions. In our next experiment we show that MACN can learn how to plan even when the state space and action space are scaled up.

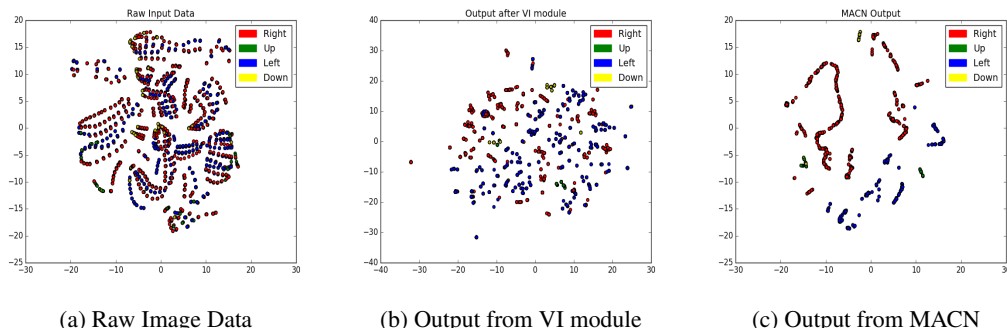

(a) Raw Image Data      (b) Output from VI module      (c) Output from MACN

Figure 6: **T-sne visualization on 2D grid worlds with tunnels**. **a**) T-sne visualization of the raw images fed into the network. Most of the input images for going into the tunnel and exiting the tunnel are the same but have different action labels. **b**) T-sne visualization from the outputs of the pre planning module. While it has done some separation, it is still not completely separable. **c**) Final output from the MACN. The actions are now clearly separable.

## 3.3 GENERAL GRAPH SEARCH

In our earlier experiments, the state space was constrained in two dimensions, and only four actions were available. It is nearly impossible to constrain every real world task to a two dimensional space with only four actions. However, it is easier to formulate a lot of partially observable planning problems as a graph.We define our environment as an undirected graph $G = (V, E)$ where the connections between the nodes are generated randomly (see Fig. 7). In Fig 7 the blue node is the start state and the red node is the goal state. Each node represents a possible state the agent could be in. The agent can only observe all edges connected to the node it currently is in thus making it partially observable. The action space for this state is then any of the possible nodes that the agent can visit next. As before, the agent only gets a reward when it reaches the goal. We also add in random start and goal positions. In addition, we add a transition probability of $0.8$. (For training details and generation of graph see Appendix.) We present our results in Table 3. On graphs with small number of nodes, the reinforcement learning with DQN and A3C sometimes converge to the optimal goal due to the small

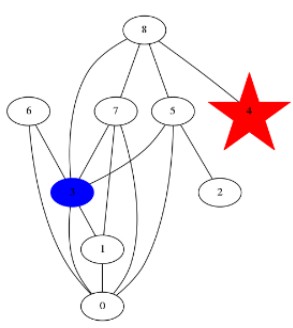

Figure 7: **9 Node Graph Search.** Blue is start and Red is goal.

state size and random actions leading to the goal node in some of the cases. However, as before the MACN outperforms all other models. On map sizes larger than 36 nodes, performance of our network starts to degrade. Further, we observe that even though the agent outputs a wrong action at some times, it still manages to get to the goal in a reasonably small number of attempts. From these results, we can conclude that MACN can learn to plan in more general problems where the state space is not limited to two dimensions and the action space is not limited to four actions.

## 3.4 CONTINUOUS CONTROL DOMAIN

Learning how to navigate in unknown environments, where only some part of the environment is observable is a problem highly relevant in robotics. Traditional robotics solve this problem by creating and storing a representation of the entire environment. However, this can quickly get memory intensive. In this experiment we extend MACN to a SE2 robot. The SE2 notation implies that the

| Model | Test Error, Success(%) | | | |
|---|---|---|---|---|
| | **9 Nodes** | **16 Nodes** | **25 Nodes** | **36 Nodes** |
| VIN | 0.57, 23.39 | 0.61, 14 | 0.68, 0 | 0.71, 0 |
| A3C | NA, 10 | NA, 7 | NA, 0 | NA, 0 |
| DQN | NA, 12 | NA, 5.2 | NA, 0 | NA,0 |
| CNN + Memory | 0.25, 81.5 | 0.32, 63 | 0.56, 19 | 0.68, 9.7 |
| MACN (LSTM) | 0.14, 98 | 0.19, 96.27 | 0.26, 84.33 | 0.29, 78 |
| MACN | **0.1, 100** | **0.18, 100** | **0.22, 95.5** | **0.28, 89.4** |

Table 3: **Performance on General Graph Search**. Test error is not applicable for the reinforcement learning models A3C and DQN

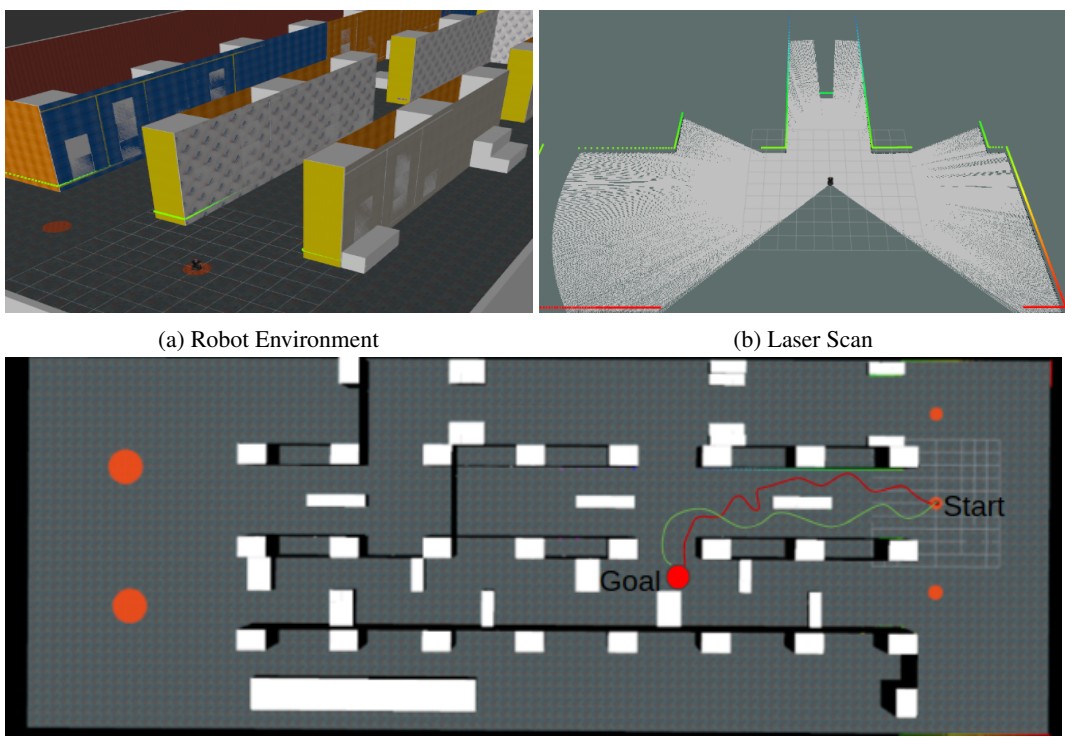

(a) Robot Environment                                    (b) Laser Scan

(c) Top Down View of Environment

Figure 8: **Navigation in a 3D environment on a continuous control robot**. **a**) The robot is spawned in a 3d simulated environment. **b**) Only a small portion of the entire map is visible at any given point to the robot **c**) The green line denotes ground truth and red line indicates the output of MACN.

robot is capable of translating in the x-y plane and has orientation. The robot has a differential drive controller that outputs continuous control values. The robot is spawned in the environment shown in Fig (8a). As before, the robot only sees a small part of the environment at any given time. In this case the robot has a laser scanner that is used to perceive the environment.

It is easy to convert this environment to a 2D framework that the MACN needs. We fix the size of the environment to a $m \times n$ grid. This translates to a $m \times n$ matrix that is fed into the MACN. The parts of the map that lie within the range of the laser scanner are converted to obstacle free and obstacle occupied regions and added to the matrix. Lastly, an additional reward map denoting a high value for the goal location and zero elsewhere as explained before is appended to the matrix and fed into the MACN. The network output is used to generate way points that are sent to the underlying controller. The training set is generated by randomizing the spawn and goal locations and using a suitable heuristic. The performance is tested on a held out test set of start and goal locations. More experimental details are outlined in the appendix.

| Model | Success (%) |
|---|---|
| DQN,A3C | 0 |
| VIN | 57.60 |
| CNN + Memory | 59.74 |
| MACN | **71.3** |

Table 4: **Performance on robot world**

We observe in Table 4 that the proposed architecture is able to find its way to the goal a large number of times and its trajectory is close to the ground truth. This task is more complex than the grid world navigation due to the addition of orientation. The lack of explicit planning in the CNN + Memory architecture hampers its ability to get to the goal in this task. In addition to this, as observed before deep reinforcement learning is unable to converge to the goal. We also report some additional results in Fig 9. In Fig 9a we show that MACN converges faster to the goal than other baselines.

In addition to rate of convergence, one of the biggest advantages of MACN over other architectures, for a fixed memory size is its ability to scale up when the size of the environment increases. We show that MACN is able to beat other baselines when scaling up the environment. In this scenario, scaling up refers to placing the goal further away from the start position. While the success percentage gradually drops to a low value, it is observed that when the memory is increased accordingly, the success percentage increases again. Lastly, in Fig 10 we observe that in the robot world, the performance of MACN scales up to goal positions further away by adjusting the size of the external memory in the differentiable block accordingly.

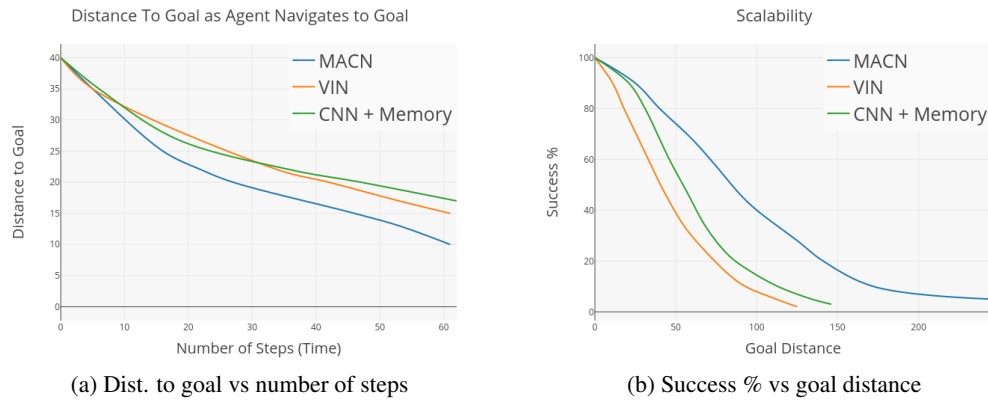

(a) Dist. to goal vs number of steps        (b) Success % vs goal distance

Figure 9: **Performance on simulated environment**. **a**) We report a plot of the number of steps left to the goal as the agent executes the learned policy in the environment (Lower is better). In this plot, the agent always starts at a position 40 steps away from the goal. **b**) The biggest advantage of MACN over other architectures is its ability to scale. We observe that as the distance to the goal increases, MACN still beats other baselines at computing a trajectory to the goal.(Higher success % is better)

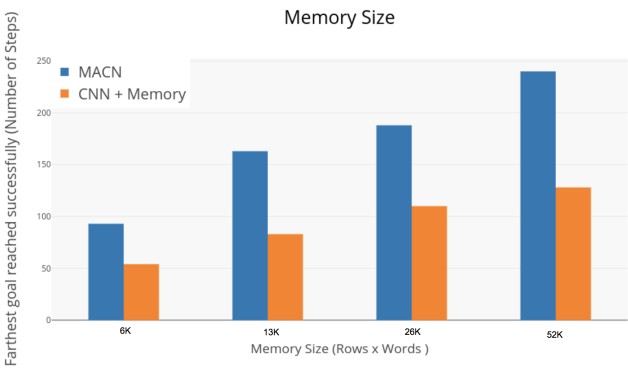

Figure 10: **Effect of Memory in Robot World** MACN scales well to larger environments in the robot world when memory is increased suitably.

### 3.5 COMPARISON WITH ANALYTICAL MOTION PLANNING BASELINES

In this section, we analyze the performance of the proposed network against traditional motion planning baselines. As stated before, for the grid world environments and the tunnel task, we obtain expert trajectories by running $A^*$ on the environment. In the case of the continuous control domain, we use the the Human Friendly Navigation (HFN) paradigm Guzzi et al. (2013) which uses a variant of $A^*$ along with a constraint for safe distances from obstacles to plan paths from start location to goal location. For the grid worlds (both with simple obstacles and local minima), we compute the ratio of path length predicted by network architecture to the path length computed by $A^*$. Our results are presented in Table 5.

The VIN alone is unable to reach the goal in a fixed number of steps. This behavior is consistent across all grid worlds. In the case of the tunnels, the VIN gets stuck inside the local minima and is unable to navigate to the goal. Thus, the ratio of path length produced by VIN to the path length produced by $A^*$ is infinite. In the case of the CNN+Memory, the network is able to navigate to the goal only when the grid world is small enough. In the case of the tunnels, the CNN+Memory learns to turn around at a fixed distance instead of exploring the tunnel all the way to the end. For example, when trained on tunnels of length 20 units and tested on tunnels of length 32 and 64 units, the CNN+Memory turns around after it has traversed 20 units in the tunnel. For this task, to demonstrate the ineffectiveness of the CNN+Memory model, we placed the goal just inside the tunnel at the dead end. Thus, the ratio of path length produced by CNN+Memory to $A^*$ is $\infty$ since the agent never explored the tunnel all the way to the end. For the case of the MACN, we observe performance close to $A^*$ for the small worlds. The performance gets worse when the size of the grid world is increased. However, the dropoff for MACN with the DNC is lesser than that of the MACN with LSTM. For the tunnel world environment, both network architectures are successfully able to emulate the performance of $A^*$ and explore the tunnel all the way to the end.

It is important to note here that $A^*$ is a model based approach and requires complete knowledge of the cost and other parameters such as the dynamics of the agent (transition probabilities). In addition, planners like $A^*$ require the user to explicitly construct a map as input, while MACN learns to construct a map to plan on which leads to more compact representations that only includes vital parts of the map (like the end of the tunnel in the grid world case). Our proposed method is a model free approach that learns

1. A dynamics model of the agent,
2. A compact map representation,
3. How to plan on the learned model and map.

This model free paradigm also allows us to move to different environments with a previously trained policy and be able to perform well by fine-tuning it to learn new features.

| Model | $A^*$ Ratio | | | | |
|---|---|---|---|---|---|
| | G(16 × 16) | G(32 × 32) | G(64 × 64) | T(L =32) | T(L = 64) |
| VIN | $\infty$ | $\infty$ | $\infty$ | $\infty$ | $\infty$ |
| CNN + Memory | 1.43 | 2.86 | $\infty$ | $\infty$ | $\infty$ |
| MACN (LSTM) | 1.2 | 1.4 | 1.62 | 1.0 | 1.0 |
| **MACN** | **1.07** | **1.11** | **1.47** | **1.0** | **1.0** |

Table 5: **Comparison to $A^*$.** G corresponds to grid world with simple obstacles with the size of the world specified inside the parenthesis. L corresponds to grid worlds with local minima/tunnels with the length of the tunnel specified inside the parenthesis. All ratios are computed during testing. For the worlds with tunnels, the network is trained on tunnels of length 20 units.

## 4 RELATED WORK

Using value iteration networks augmented with memory has been explored before in (Gupta et al., 2017). In their work a planning module together with a map representation of a robot's free space is

used for navigation in a partially observable environment using image scans. The image scans are projected into a 2D grid world by approximating all possible robot poses. This projection is also learned by the model. This is in contrast to our work here in which we design a general memory based network that can be used for any partially observed planning problem. An additional difference between our work and that of (Gupta et al., 2017) is that we do not attempt to build a 2D map of the environment as this hampers the ability of the network to be applied to environments that cannot be projected into such a 2D environment. We instead focusing on learning a belief over the environment and storing this belief in the differentiable memory. Another similar work is that of (Oh et al., 2016) where a network is designed to play Minecraft. The game environment is projected into a 2D grid world and the agent is trained by RL to navigate to the goal. That network architecture uses a CNN to extract high level features followed by a differentiable memory scheme. This is in contrast to our paper where we approach this planning by splitting the problem into local and global planning. Using differential network schemes with CNNs for feature extraction has also been explored in (Chen et al., 2017). Lastly, a recently released paper Neural SLAM (Zhang et al., 2017) uses the soft attention based addressing in DNC to mimic subroutines of simultaneous localization and mapping. This approach helps in exploring the environment robustly when compared to other traditional methods. A possible extension of our work presented here, is to use this modified memory scheme with the differentiable planner to learn optimal paths in addition to efficient exploration. We leave this for future work.

## 5 CONCLUSION

Planning in environments that are partially observable and have sparse rewards with deep learning has not received a lot of attention. Also, the ability of policies learned with deep RL to generalize to new environments is often not investigated. In this work we take a step toward designing architectures that compute optimal policies even when the rewards are sparse, and thoroughly investigate the generalization power of the learned policy. In addition we show our network is able to scale well to large dimensional spaces.

The grid world experiments offer conclusive evidence about the ability of our network to learn how to plan in such environments. We address the concern of oversimplifying our environment to a 2D grid world by experimenting with planning in a graph with no constraint on the state space or the action space. We also show our model is capable of learning how to plan under continuous control. In the future, we intend to extend our policies trained in simulation to a real world platform such as a robot learning to plan in partially observable environments. Additionally, in our work we use simple perfect sensors and do not take into account sensor effects such as occlusion, noise which could aversely affect performance of the agent. This need for perfect labeling is currently a limitation of our work and as such cannot be applied directly to a scenario where a sensor cannot provide direct information about nearby states such as a RGB camera. We intend to explore this problem space in the future, where one might have to learn sensor models in addition to learning how to plan.

## ACKNOWLEDGEMENT

We gratefully acknowledge the support of ARL grants W911NF-08-2-0004 and W911NF-10-2-0016, ARO grant W911NF-13-1-0350, ONR grants N00014-07-1-0829, N00014-14-1-0510, DARPA grant HR001151626/HR0011516850 and DARPA HR0011-15-C-0100, USDA grant 2015-67021-23857 NSF grants IIS-1138847, IIS-1426840, CNS-1446592, CNS-1521617, and IIS-1328805. Clark Zhang is supported by the National Science Foundation Graduate Research Fellowship Program under Grant No. DGE-1321851. The authors would like to thank Anand Rajaraman, Steven Chen, Jnaneswar Das, Ekaterina Tolstoya and others at the GRASP lab for interesting discussions related to this paper.

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

**APPENDIX**

## A    GRID WORLD EXPERIMENTS

For the grid world we define our sensor to be a $7 \times 7$ patch with the agent at the center of this patch. Our input image $I$ to the VI module is $[m \times n \times 2]$ image where $m,n$ are the height and width of the image. $I[:,:,0]$ is the sensor input. Since we set our rewards to be sparse, $I[:,:,1]$ is the reward map and is zero everywhere except at the goal position $(m_{goal}, n_{goal})$. $I$ is first convolved to obtain a reward image $R$ of dimension $[n \times m \times u]$ where $u$ is the number of hidden units (vary between 100-150). This reward image is sent to the VI module. The value maps from the VI module after K iterations are fed into the memory network controller. The output from the network controller (here a LSTM with 256 hidden units) and the access module is concatenated and sent through a linear layer followed by a soft max to get a probability distribution over $\mathcal{A}$.

During training and testing we roll out our sequence state by state based on the ground truth or the networks output respectively. Further, the set of transitions from start to goal to are considered to be an episode. During training at the end of each episode the internal state of the controller and the memory is cleared. The size of the external memory is $32 \times 8$ the grid world task. An additional hyperparameter is the number of read heads and write heads. This parameter controls the frequency of reads vs frequency of writes. For the the grid world task, we fix the number of read heads to 4 and the number of write heads to 1. For the grid world with simple obstacles, we observe that the MACN performs better when trained with curriculum (Bengio et al., 2009). This is expected since both the original VIN paper and the DNC paper show that better results are achieved when trained with curriculum. For establishing baselines, the VIN and the CNN+Memory models are also trained with curriculum learning. In the grid world environment it is easy to define tasks that are harder than other tasks to aid with curriculum training. For a grid world with size $(m, n)$ we increase the difficulty of the task by increasing the number of obstacles and the maximum size of the obstacles. Thus, for a $32 \times 32$ grid, we start with a maximum of 2 obstacles and the maximum size being $2 \times 2$. Both parameters are then increased gradually. The optimal action in the grid world experiments is generated by A star (Russell & Norvig, 2003). We use the Manhattan distance between our current position and the goal as a heuristic. Our error curves on the test set for the MACN with the LSTM and the addressable memory scheme are shown in Figure 11.

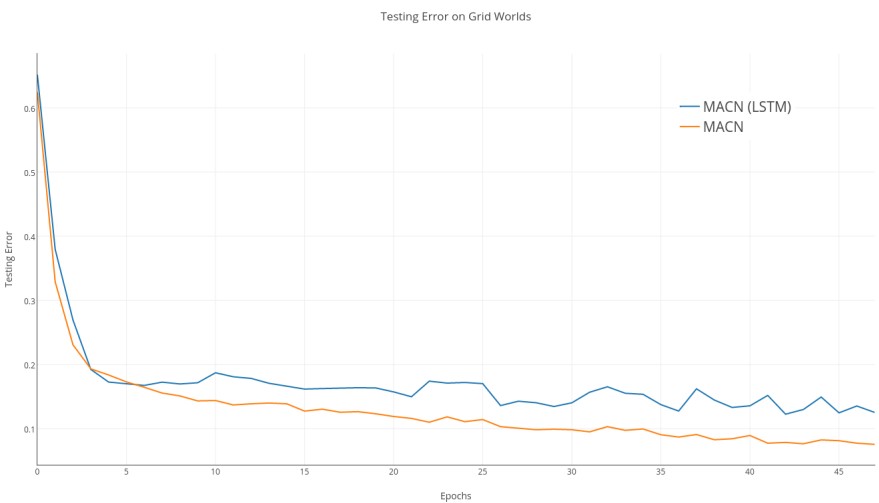

Figure 11: Testing Error on grid world (32 x 32)

## B  ENVIRONMENTS WITH TUNNELS

We setup the network the same way as we did for the grid world experiments with blob shaped obstacles. Due to the relatively simple structure of the environment, we observe that we do not really need to train our networks with curriculum. Additionally, the read and write heads are both set to 1 for this experiment.

### B.1  VI MODULE + PARTIAL MAPS

We observe that for the tunnel shaped obstacle when just the VI module is fed the partial map (stitched together version of states explored) as opposed to the sensor input, it performs extremely well and is able to generalize to new maps with longer tunnels without needing any memory. This is interesting because it proves our intuition about the planning task needing memory. Ideally we would like the network to learn this partial map on its own instead of providing it with a hand engineered version of it. The partial map represents an account of all states visited in the past. We argue that not all information from the past is necessary and the non redundant information that is required for planning in the global environment can be learned by the network. This can be seen in the memory ablation.

### B.2  DQN

As stated in the main paper, we observe that the DQN performs very poorly since the rewards are very sparse. The network is setup exactly as described in  (Mnih et al., 2013a). We observe that even after 1 million iterations, the agent never reaches the goal and instead converges to a bad policy. This can be seen in Figure 12. It is clear that under the random initial policy the agent is unable to reach the goal and converged to a bad policy. Similar results are observed for A3C. Further, it is observed that even when the partial map instead of the sensor input is fed in to DQN, the agent does not converge.

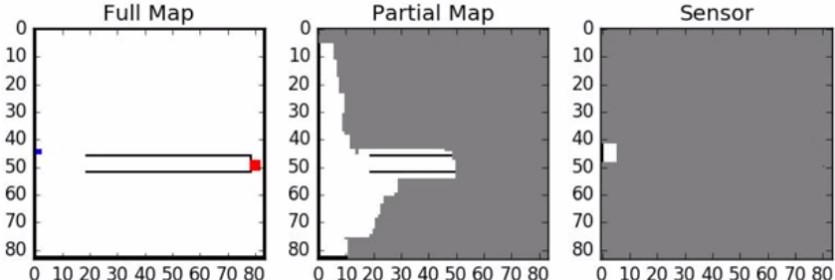

Figure 12: **Learning optimal policy with Deep Q networks.** In this case the input to the DQN is the sensor input. State after 1 million iterations. The agent gets stuck along the wall (left wall between 40 and 50)

### B.3  VISUALIZING THE MEMORY STATE

For the tunnel task, we use an external memory with 32 rows ($N = 32$) and a word size of 8 ($W = 8$). We observe that when testing the network, the memory registers a change in its states only when important events are observed. In Figure 11, the left hand image represents the activations from the memory when the agent is going into the tunnel. *We observe that the activations from the memory remain constant until the agent observes the end of the tunnel*. The memory states change when the agent observes the end of the tunnel, when it exits the tunnel and when it turns towards its goal (Figure 13). Another key observation for this task is that the MACN is prone to over fitting for this task. This is expected because ideally, only three states need to be stored in the memory; entered the tunnel, observe end of tunnel and exit tunnel. To avoid overfitting we add L2 regularization to our memory access weights.

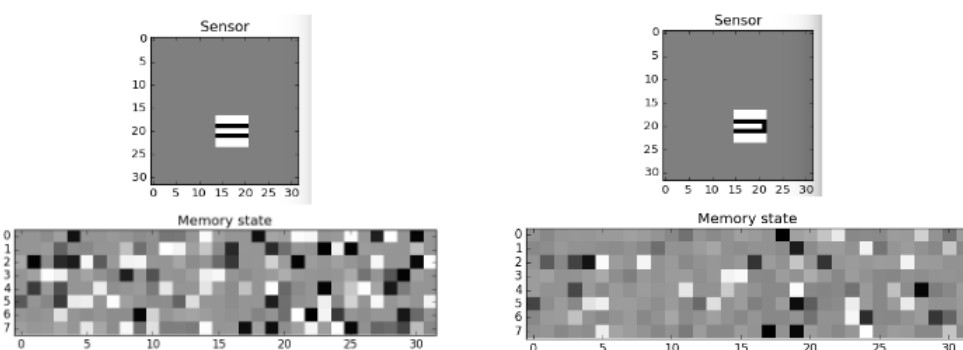

Figure 13: Shift in memory states just before and after the end of the tunnel is observed. Once the agent has turned around, the memory state stays constant till it reaches the end of the tunnel.

## C  GRAPH EXPERIMENTS

For the graph experiment, we generate a random connected undirected graph with $N$ nodes. We will call this graph $G = (V, E)$, with nodes $V = \{V_1, V_2, \ldots, V_N\}$ and edges, $E$. The agent, at any point in the simulation, is located at a specific node $V_i$ and travels between nodes via the edges. The agent can take actions from a set $U = \{u_1, u_2, \ldots, u_N\}$ where choosing action $u_i$ will attempt to move to node $V_i$. We have transition probabilities

$$p(V_j | V_i, u_k) = \begin{cases} 0, \text{if } k \neq j \text{ or } (V_i, V_j) \notin E \\ 1, \text{if } k = j \text{ and } (V_i, V_j) \in E \end{cases} \quad (5)$$

At each node, the agent has access to the unique node number (all nodes are labeled with a unique ID), as well as the $(A)_i$, the $i^{th}$ row of the adjacency matrix $A$. It also has access to the unique node number of the goal (but no additional information about the goal).

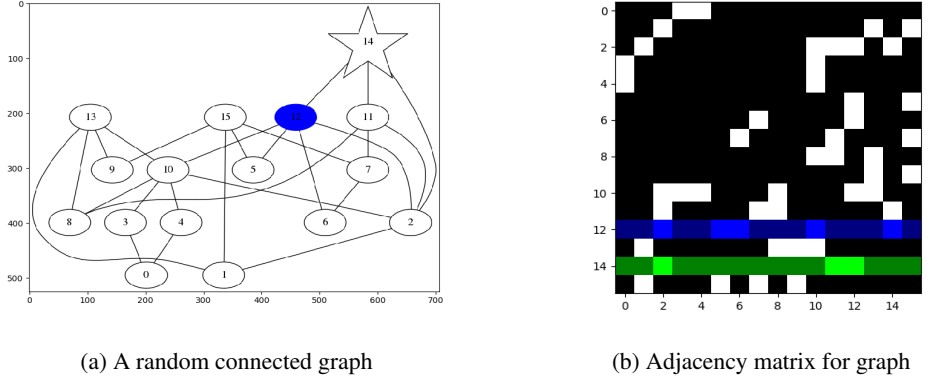

(a) A random connected graph

(b) Adjacency matrix for graph

Figure 14: **a**) A random connected undirected graph with a the goal given by the star-shaped node, and the current state given by the blue node **b**) Shows the corresponding adjacency matrix where white indicates a connection and black indicates no connection. The goal is given by the row shaded in green, and the current state is given by the row shaded in blue.

To train the network to navigate this graph, we used supervised training with an expert demonstrating an intended behavior (breadth first search). Training samples were generated by running breadth first search (and connecting nodes that are explored by traveling previously explored nodes of the graph). Thus, for each state of the node and goal, we obtain a desired action. To fit this into the framework of our network and 2D convolutions, we reshaped the row vector of the matrix into a matrix that could

use the same convolution operation. The reward prior is also a row vector with a 1 at the index of the goal node and zero everywhere else. This row vector is reshaped and stacked with the observation. We train the graph by giving example paths between pairs of nodes. We then test on pairs of nodes not shown during training. The training network is setup as before in the grid world navigation task. Due to the increased action space and state space, this task is significantly more complex than the grid world navigation task. We train MACN and the baselines with curriculum training. In the graph task it is easy to define a measure of increasing complexity by changing the number of hops between the start state and the goal state. Additionally, for the graph task the number of read heads and write heads are set to 1 and 4 respectively.

## D  CONTINUOUS CONTROL

Navigating an unknown environment is a highly relevant problem in robotics. The traditional methodology localizes the position of the robot in the unknown world and tries to estimate a map. This approach is called Simultaneous Localization and Mapping (SLAM) and has been explored in depth in robotics (Thrun & Leonard, 2008). For the continuous control experiment, we use a differential drive robot (Figure 15). The robot is equipped with a head mounted LIDAR and also has a ASUS Xtion Pro that can provide the depth as well as the image from the front facing camera. In this work, we only use the information from the LIDAR and leave the idea of using data from the camera for future work. The ground truth maps are generated by using

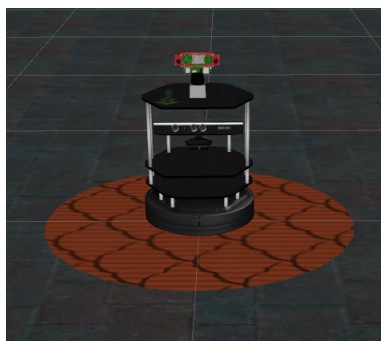

Figure 15: **Simulated ground robot**

Human Friendly Navigation (HFN) (Guzzi et al., 2013) which generates reactive collision avoidance paths to the goal. Given a start and goal position, the HFN algorithm generates way points that are sent to the controller. For our experiment, we generate a tuple of $(x, y, \theta)$ associated with every observation. To train the network, a $m \times n$ matrix (environment matrix) corresponding to the $m \times n$ environment is initialized. A corresponding reward array (reward matrix) also of size $m \times n$ with a 1 at the goal position and zero elsewhere is concatenated with the environment matrix. The observations corresponding to the laser scan are converted to a $j \times k$ matrix (observation matrix) where $j < m$ and $k < n$. The values at the indices in the environment array corresponding to the local observation are updated with the values from the observation matrix. At every iteration, the environment matrix is reset to zero to ensure that the MACN only has access to the partially observable environment.

For the continuous world we define our observation to be a $10 \times 10$ matrix with the agent at the bottom of this patch. We change our formulation in the previous cases where our agent was at the center since the LIDAR only has a 270 degree field of view and the environment behind the robot is not observed. Our input image $I$ to the VI module is $[m \times n \times 2]$ image where $m = 200$,$n = 200$ are the height and width of the environment. $I[:, :, 0]$ is the sensor input. $I$ is first convolved to obtain a reward image $R$ of dimension $[n \times m \times u]$ where $u$ is the number of hidden units (200 in this case). The K (parameter corresponding to number of iterations of value iteration) here is 40. The network controller is a LSTM with 512 hidden units and the external memory has 1024 rows and a word size of 512. We use 16 write heads and 4 read heads in the access module. The output from the access module is concatenated with the output from the LSTM controller and sent through a linear layer followed by a soft max to get probability distributions for $(x, y, \theta)$. We sample from these distributions to get the next waypoint. These way points are then sent to the controller. The waypoints are clipped to ensure that the robot takes incremental steps.

For this task, we find that the performance increases when trained by curriculum training. MACN in addition to the baselines is first trained on maps where the goal is close and later trained on maps where the goal is further away. An additional point here, is that due to the complexity of the task, we train and test on the same map. Maps in the train set and test set differ by having random start and goal regions.

