# OpenReview forum: "Memory Augmented Control Networks"
_ICLR.cc/2018/Conference — Accept (Poster)_

### Official Review · AnonReviewer1 · 2017-11-22
**What about strong motion-planning baselines?**

**Rating:** 4
**Confidence:** 5

**Review:**

Summary:

A method is proposed for robot navigation in partially observable scenarios. E.g. 2D navigation in a grid world from start to goal but the robot can only sense obstacles in a certain radius around it. A learning-based method is proposed here which takes the currently discovered partial map as input to convolutional layers and then passes through K-iterations of a VIN module to a final controller. The controller takes as input both the convolutional features, the VIN module and has access to a differential memory module. A linear layer takes inputs from both the controller and memory and predicts the next step of the robot. This architecture is termed as MACN.

In experiments on 2D randomly generated grid worlds, general graphs and a simulated ground robot with a lidar, it is shown that memory is important for navigating partially observable environments and that the VIN module is important to the architecture since a CNN replacement doesn't perform as well. Also larger start-goal distances can be better handled by increasing the memory available.

Comments:

- My main concern is that there are no non-learning based obvious baselines like A*, D*, D*-Lite and related motion planners which have been used for this exact task very successfully and run on real-world robots like the Mars rover. In comparison to the size of problems that can be handled by such planners the experiments shown here are much smaller and crucially the network can output actions which collide with obstacles while the search-based planners by definition will always produce feasible paths and require no training data. I would like to see in the experimental tables, comparison to path lengths produced by MACN vs. those produced by D*-Lite or Multi-Heuristic A*. While it is true that motion-planning will keep the entire discovered map in memory for the problem sizes shown here (2D maps: 16x16, 32x32, 64x64 bitmaps, general graphs: 9, 16, 25, 36 nodes) that is on the order of a few kB memory only. For the 3D simulated robot which is actually still treated as a 2D task due to the line lidar scanner MxN bitmap is not specified but even a few Mb is easily handled by modern day embedded systems. I can see that perhaps when map sizes exceed say tens of Gbs then perhaps MACN's memory will be smaller to obtain similar performance since it may learn better map compression to better utilize the smaller budget available to it. But experiments at that scale have not been shown currently.

- Figure 1: There is no sensor (lidar or camera or kinect or radar) which can produce the kind of sensor observations shown in 1(b) since they can't look beyond occlusions. So such observations are pretty unrealistic.

- "The parts of the map that lie within the range of the laser scanner are converted to obstacle-free ...": How are occluded regions marked?

---

> ### Author Response · Authors · 2017-12-12
> **Official Reply to AnonReviewer1**
>
> Dear Reviewer,
>
> We would like to thank you for your detailed review. We would like to answer some of the points raised by you in our response here :
>
> We have tried to address the reviewers concerns about comparing with motion planning baselines by adding another subsection. We agree it is useful to compare to A* and have added comparisons. We request the reviewer to look at Section 3.5 in the latest version.
>
> That being said, we would like to point out several things here :
>
> 1) There is a key difference between our approach and using a planning algorithm such as A*. A* is a model-based approach where you need to explicitly know beforehand knowledge about the cost, transition probabilities of the agent and explicitly construct a map. The motivation behind using a model-free approach such as ours is that these transition probabilities and cost function (in our case reward map -> low reward near obstacles and high reward in open space) is learned by the agent.
>
> This is, in fact, the biggest motivation for using end to end learning approaches!  One does not need to explicitly know the model beforehand and can use the neural network to approximate it. Our proposed model learns the transition probabilities the cost map and the learns how to plan on the local map.
>
> 2) Another advantage of using a model-free approach such as ours as opposed to A* is that our model learns a compact representation of the environment. This can be seen in Experiment 2 (grid world with tunnels)  and Section B in the Appendix. In the case of the tunnel environment, we can make the tunnels arbitrarily long (say 500 units in length). A* would have to expand all nodes going into the tunnel and would need to remember the entire map.
> However with our approach, we can use the same memory size for both 20 length tunnels as well as 500 since it records only the events where we enter and exit the tunnel as well as the end of the tunnel. Further, these events were not hand engineered. Instead, the network learned what events were important to understand the topology of a tunnel.  '
>
> We absolutely agree with the reviewer that our sensor models are simplistic and we assume perfect models. In this work, we are focused on learning how to navigate a partially observable environment when an architecture consisting of a differentiable planner and memory are used. In future work, we would focus on extending our work to model sensor effects such as noise, occlusions.
>
> We hope this answers some of your concerns about our paper and you reconsider our paper more favorably.

---

### Official Review · AnonReviewer3 · 2017-11-28
**The paper proposes a novel neural network architecture for planning in partially observable environments with sparse rewards. It uses a differentiable memory module to maintain an estimate of the geometry of the partially observable state, and splits planning into a two-level hierarchical process. Experiments in several domains demonstrate the validity of the proposed architecture.**

**Rating:** 6
**Confidence:** 2

**Review:**

The paper addresses the important problem of planning in partially observable environments with sparse rewards, and the empirical verification over several domains is convincing. My main concern is that the structure of these domains is very similar - essentially, a graph where only neighboring vertices are directly observable, and because of this, the proposed architecture might not be applicable to planning in general POMDPs (or, in their continuous counterparts, state-space models). The authors claim that what is remembered by the planner does not take the form of a map, but isn't the map estimate \hat{m} introduced at the end of Section 2.1 precisely such a map? From Section 2.4, it appears that these map estimates are essential in computing the low-level policies from which the final, high-level policy is computed. If the ability to maintain and use such local maps is essential for this method, its applicability is likely restricted to this specific geometric structure of domains and their observability.

Some additional comments:

P. 2, Section 2.1: does H(s) contain 0s for non-observable and 1s for observable states? If yes, please state it.

P. 3: the concatenation of state and observation histories is missing from the definition of the transition function.

P. 3, Eq. 1: overloaded notation - if T is the transition function for the large MDP on histories, it should not be used for the transition function between states. Maybe the authors meant to use f() for that transition?

P. 3, Eq. 3: the sum is over i, but it is not clear what i indexes.

P.3, end of Section 2.1: when computing the map estimate \hat{m}, shouldn't the operator be min, that is, a state is assumed to be open (0), unless one or more observations show that it is blocked (-1)?

P.5: the description of the reward function is inconsistent - is it 0 at the goal state, or >0?

P. 11, above Fig. 9: typo, "we observe that the in the robot world"

---

> ### Author Response · Authors · 2017-12-03
> **Official Reply to AnonReviewer3**
>
> Dear Reviewer,
>
> Thank you for your detailed review. We would like to answer some of the points raised by you in our response here :
>
> 1) We agree that on a first pass it might look like the structure of the domains looks very similar. However, while writing this paper our focus was on the environments/domains that one might encounter in robotics and real world applications. We choose to demonstrate the feasibility of our architecture in such 2D/2.5D worlds and look to answer problems faced by other learning architectures in such worlds. Further, we added the graph experiment in section 3.3 to break up some of the structural similarity between the domains.  The graph experiment differs from the other domains in that the state space is no longer 2 dimensional. Further, the number of states observed by the agent and the number of valid actions varies as the agent visits each node. This is because the action space now depends on the number of vertices connected to the current node that the agent is in. Additionally, the action space in these graphs is also no longer a choice between up/down/left/right. The agent has to learn to pick the correct next node to visit and there are N-1 choices (where N is the number of nodes).
>
>
> 2) We completely agree with the reviewer that remembering a map might hamper the ability of the work to be extended to other domains which might not have explicit geometric structure. This is in fact one of the limitations of "Cognitive Mapping and Planning for Visual Navigation" by Gupta et. al where learning an explicit top down map of the 3d environment might not be possible in some domains. Instead, in our work, we maintain a belief estimate over the environment which is represented in the external memory as a set of activations. We would like to draw the reviewer's attention to Fig 13 in the appendix. In this figure, we show the map estimate stored in the memory for the tunnel task. As one can see, the information stored in the memory does not correspond to the geometric structure of the environment. Instead, our proposed architecture learns to output different activations when the critical parts of the environment are observed by the agent. In the tunnel task, the memory exhibits one kind of activations when the agent observes the end of the tunnel, and when it turns out of the tunnel. For all other events, the memory shows no change in its activations. Additionally, when looking at the read/write weights when entering and exiting the tunnel, we see that the write weights are activated till the agent sees the end of the tunnel and the read weights are activated when the agent turns around. Thus, planning by remembering important events encountered in the environment allows us to use the proposed planner to domains where geometric structure might not exist. This is also something we wished to demonstrate by planning on graphs where there is no such explicit structure.
>
> We thank the reviewer for pointing out the typos and other potentially confusing statements and will address them in the updated version.
>
>
> We hope, this answers some of your concerns about our paper.

---

> > ### Author Response · Authors · 2017-12-12
> > **Updated Paper**
> >
> > Dear Reviewer,
> >
> > We have updated our paper to address the typos you had pointed out in our paper.
> >
> > Additionally,  would like to answer one of the questions you had raised
> >
> > "P.3, end of Section 2.1: when computing the map estimate \hat{m}, shouldn't the operator be min, that is, a state is assumed to be open (0), unless one or more observations show that it is blocked (-1)?"
> >
> > If one were to consider a 1x1 in which we have 2 observations over time. if both obs are zero, the sum is zero and the max is zero thus indicating that the state is open.
> >
> > if both obs are -1, sum is -2. The max is -1 indicating state is blocked.
> > If one of the observations is -1 and the other 0, the sum is -1 and the max is still -1 telling us that the state is blocked.
> >
> > All other typos have been addressed in the paper.
> > We hope this answers your concerns about our paper and you re consider our paper favorably.

---

### Official Review · AnonReviewer2 · 2017-11-29
**Clever idea with strong supporting experimental evidence, but paper is missing key details about the approach**

**Rating:** 9
**Confidence:** 4

**Review:**

The paper presents a method for navigating in an unknown and partially observed environment is presented. The proposed approach splits planning into two levels: 1) local planning based on the observed space and 2) a global planner which receives the local plan, observation features, and access to an addressable memory to decide on which action to select and what to write into memory.

The contribution of this work is the use of value iteration networks (VINs) for local planning on a locally observed map that is fed into a learned global controller that references history and a differential neural computer (DNC), local policy, and observation features select an action and update the memory. The core concept of learned local planner providing additional cues for a global, memory-based planner is a clever idea and the thorough analysis clearly demonstrates the benefit of the approach.

The proposed method is tested against three problems: a gridworld, a graph search, and a robot environment. In each case the proposed method is more performant than the baseline methods.  The ablation study of using LSTM instead of the DNC and the direct comparison of CNN + LSTM support the authors’ hypothesis about the benefits of the two components of their method. While the author’s compare to DRL methods with limited horizon (length 4), there is no comparison to memory-based RL techniques. Furthermore, a comparison of related memory-based visual navigation techniques on domains for which they are applicable should be considered as such an analysis would illuminate the relative performance over the overlapping portions problem domains  For example, analysis of the metric map approaches on the grid world or of MACN on their tested environments.

Prior work in visual navigation in partially observed and unknown environments have used addressable memory (e.g., Oh et al.) and used VINs (e.g., Gupta et al.) to plan as noted. In discussing these methods, the authors state that these works are not comparable as they operate strictly on discretized 2d spaces. However, it appears to the reviewer that several of these methods can be adapted to higher dimensions and be applicable at least a subclass (for the euclidean/metric map approaches) or the full class of the problems (for Oh et al.), which appears to be capable to solve non-euclidean tasks like the graph search problem. If this assessment is correct, the authors should differentiate between these approaches more thoroughly and consider empirical comparisons. The authors should further consider contrasting their approach with “Neural SLAM” by Zhang et al.

A limitation of the presented method is requirement that the observation “reveals the labeling of nearby states.” This assumption holds in each of the examples presented: the neighborhood map in the gridworld and graph examples and the lidar sensor in the robot navigation example. It would be informative for the authors to highlight this limitation and/or identify how to adapt the proposed method under weaker assumptions such as a sensor that doesn’t provide direct metric or connectivity information such as a RGB camera.

Many details of the paper are missing and should be included to clarify the approach and ensure reproducible results. The reviewer suggests providing both more details in the main section of the paper and providing the precise architecture including hyperparameters in the supplementary materials section.

---

> ### Author Response · Authors · 2017-12-26
> **Official reply to AnonReviewer2**
>
> Dear Reviewer,
>
> We would first off like to thank you for your strong support and feedback on our paper. Your detailed reviews will definitely help us in improving our paper.
>
> We would like to answer some of the points raised by you in our response here:
>
> We set up the CNN+Memory architecture to emulate the FRMQN from Oh et. al's work as closely as possible. The DNC actually improves upon the read, write and context architecture described in the paper. Further, in our experiments, we found that when training the CNN+Memory architecture with supervised learning, the network performed worse than our MACN. We hypothesized that if supervised learning was unable to learn a reasonable policy, then any reinforcement learning paradigm with sparse rewards would definitely do worse.
>
>
> We would like to thank you for bringing to our attention the work of Zhang et. al - "Neural SLAM". To the best of our understanding, this paper focuses on using a SLAM formulation in a deep reinforcement learning paradigm which helps in exploration. Exploration is one topic that we have not explored in this work since we assume that there is always a path to the goal. In future work, we intend to extend our network to be trained with reinforcement learning instead of supervised learning. In such a setting, a Neural SLAM style architecture might help with exploration when the environment presents sparse rewards.
>
> We have added a note to Section 5 regarding the need for perfect labeling of nearby states. We agree additional work is required to model sensors such as an RGB camera where such direct labeling might not be possible. The focus of this paper is to investigate the feasibility of a hierarchical learning scheme for planning in partially observable environments and hence we assume perfect sensors. In future work, using real-world sensors that do not always give a perfect labeling of nearby states will be one of our goals.
>
> We have included all details one would need to reproduce our work in Section 2.4 under the computation graph. Further, experiment specific details are included in the appendix. It might be possible to present these in a more reader friendly format such as a table in the camera-ready version of the paper if required? Additionally, we intend to make our code publicly available.

---

### Public Comment · (anonymous) · 2017-12-11
**Neural SLAM: Learning to Explore with External Memory**

You might be interested to take a look

Neural SLAM: Learning to Explore with External Memory
(https://arxiv.org/pdf/1706.09520.pdf)

We present an approach for agents to learn representations of a global map from sensor data, to aid their exploration in new environments. To achieve this, we embed procedures mimicking that of traditional simultaneous localization and mapping (SLAM) into the soft attention based addressing of external memory architectures, in which the external memory acts as an internal representation of the environment for the agent. This structure encourages the evolution of SLAMlike behaviors inside a completely differentiable deep neural network. We show that this approach can help reinforcement learning agents to successfully explore new environments where long-term memory is essential. We validate our approach in both challenging grid-world environments and preliminary Gazebo experiments. A video of our experiments can be found at: https://goo.gl/G2Vu5y.

---

> ### Author Response · Authors · 2017-12-26
> **Neural SLAM: Learning to Explore with External Memory**
>
> Thank you for bringing to our attention this work. This is definitely very interesting and we have included a pointer
> to your work in our related work section.

---

### Decision · Program_Chairs · 2018-01-29
**ICLR 2018 Conference Acceptance Decision**

**Decision:**

Accept (Poster)

**Comment:**

The authors have proposed an architecture that incorporates a VIN with a DNC to combine low level planning with high level memory-based optimization, resulting in a single policy for navigation and other similar problems that is trained end-to-end with sparse rewards. The reviews are mixed, but the authors did allay the concerns of the most negative reviewer by adding a comparison to traditional motion planning (A*) algorithms.